# Molecular Imaging of ACE2 Expression in Infectious Disease and Cancer

**DOI:** 10.3390/v15101982

**Published:** 2023-09-23

**Authors:** Zhiyao Li, Abbie Hasson, Lasya Daggumati, Hanwen Zhang, Daniel L. J. Thorek

**Affiliations:** 1Mallinckrodt Institute of Radiology, Washington University in St. Louis School of Medicine, St. Louis, MO 63110, USA; lzhiyao@wustl.edu (Z.L.); a.hasson@wustl.edu (A.H.); hanwen.zhang@wustl.edu (H.Z.); 2Program in Quantitative Molecular Therapeutics, Washington University in St. Louis School of Medicine, St. Louis, MO 63110, USA; lasya.daggumati@wustl.edu; 3Department of Biomedical Engineering, Washington University, St. Louis, MO 63110, USA; 4School of Medicine Missouri, University of Missouri-Kansas City, Kansas, MO 64108, USA; 5Siteman Cancer Center, St. Louis, MO 63110, USA

**Keywords:** ACE2, radiotracer, quantitative imaging, diagnosis, patient management

## Abstract

Angiotensin-converting enzyme 2 (ACE2) is a cell-surface receptor that plays a critical role in the pathogenesis of SARS-CoV-2 infection. Through the use of ligands engineered for the receptor, ACE2 imaging has emerged as a valuable tool for preclinical and clinical research. These can be used to visualize the expression and distribution of ACE2 in tissues and cells. A variety of techniques including optical, magnetic resonance, and nuclear medicine contrast agents have been developed and employed in the preclinical setting. Positron-emitting radiotracers for highly sensitive and quantitative tomography have also been translated in the context of SARS-CoV-2-infected and control patients. Together this information can be used to better understand the mechanisms of SARS-CoV-2 infection, the potential roles of ACE2 in homeostasis and disease, and to identify potential therapeutic modulators in infectious disease and cancer. This review summarizes the tools and techniques to detect and delineate ACE2 in this rapidly expanding field.

## 1. Introduction

Angiotensin-converting enzyme 2 (ACE2) is a transmembrane protein that has gained significant attention in the fields of infectious disease and physiology. ACE2 belongs to the carboxypeptidase family and is predominantly expressed on the surface of cells in various tissues, including the lungs, heart, kidneys, and intestines. It acts as a key component of the renin–angiotensin–aldosterone system (RAAS) and plays a crucial role in regulating blood pressure and fluid balance [1]. ACE2 counteracts the actions of the angiotensin-converting enzyme (ACE) by converting angiotensin II into angiotensin-(1-7), which promotes vasodilation and exhibits anti-inflammatory and antifibrotic effects [2]. Beyond its physiological role, ACE2 gained widespread attention due to its involvement in the cellular entry of coronaviruses.

The obligatory roles of angiotensin-converting enzyme 2 in normal organ function and in disease are mediated by tissue- and cell-specific profiles and the magnitude of expression. Direct sampling of all tissues is cumbersome and invasive, and impossible in the clinical setting. Methods that enable imaging of the distribution and the expression level of ACE2 provide crucial information about these biological processes. In this review, we describe methods and results for imaging ACE2 in relation to COVID-19 infection; in tumor biology; and in pathogenesis of liver diseases. By visualizing ACE2 expression and distribution, we enhance our understanding of viral pathogenesis, monitor its presence in different tissues, and develop targeted therapeutic approaches. This information offers insights into the role of ACE2 in disease progression, a way to identify potential biomarkers, and a means to monitor methods to improve patient outcomes. Visualizing ACE2 expression in vivo throughout the course of disease development may be significant for pathologies, including COVID-19 and cancer, which can have long-term effects. These longitudinal data may lead to improved individualized care for patients and detect at-risk populations for certain diseases.

We pay particular attention to the application of ACE2 imaging in the context of COVID-19, as it has been demonstrated that it is through ACE2 that the SARS-CoV-2 virus enters and infects human cells, leading to a multi-organ impact. The spike proteins of these viruses bind to ACE2, facilitating viral entry into host cells [3]. Thus, ACE2 is opening novel avenues for our understanding of COVID-19. The interaction of several coronaviruses with ACE2 has significant implications for viral tropism, pathogenesis, and potential therapeutic interventions. The majority of animal models used to study infectious disease, and COVID-19, are mouse in origin; however, murine ACE2 does not bind the SARS-CoV-2 spike (S) protein. Therefore, transgenic human-ACE2 mice have been used in preclinical imaging agent development, and in tumor xenograft models [4,5]. For the latter, abnormal ACE2 expression is implicated in tumor growth and progression in certain cancers [6], as well as in liver disease [7]. Improved understanding of the regulatory role of ACE2 and its expression in different disease states through imaging provides useful information to guide treatment.

To date, several imaging modalities, including fluorescence imaging, magnetic resonance (MR) imaging, and nuclear medicine imaging, have been employed for ACE2 visualization in preclinical and clinical settings. Methods relying on nuclear imaging technologies have been the most widely studied and clinically translated, because of its high sensitivity to provide non-invasive and quantitative assessment of the systemic distribution of ACE2 in various diseases through the ACE2-targeting radiotracers.

## 2. Background on COVID-19

COVID-19 is a highly contagious infectious disease caused by SARS-CoV-2. Since it was first reported in Wuhan, Hubei Province, China in December 2019, it has spread worldwide with disastrous effects. In March 2020, the World Health Organization (WHO) declared it a global pandemic. COVID-19 is a positive single-stranded RNA virus (+ssRNA). Structurally, the virus is mainly composed of four main structural proteins: spike, envelope glycoprotein, nucleocapsid, and membrane protein. The COVID-19 virus primarily enters human cells by binding to ACE2 receptors on the surfaces of host cells [3]. The spike protein on the surface of the virus facilitates this binding process, leading to cellular entry and subsequent viral replication [8,9]. The downregulation of ACE2 receptors caused by the virus can disrupt the balance of the renin–angiotensin system, contributing to the pathogenesis of COVID-19.

For the majority, the mild symptoms of COVID-19 primarily manifest in the upper respiratory tract and digestive system. In severe cases, the illness progresses rapidly to acute respiratory failure, septic shock, blood clotting, and life-threatening complications. Fortunately, most patients have asymptomatic disease or quickly recover, while the elderly and those with chronic underlying diseases have a poor prognosis [10]. While ongoing, the long-term follow-up of recovered patients reveals that a significant proportion (with some estimates exceeding 10%) suffer from long-term symptoms post-COVID-19. These may present as cardiovascular disease, shortness of breath, loss of smell and taste, loss of mental acuity, and issues with reproductive function, among others [11].

In the detection of COVID-19, the gold standard is molecular detection using real-time PCR detection of SARS-CoV-2 nucleic acid in nasopharyngeal swabs. Clinical imaging has played a role in monitoring patient symptoms. For example, planar X-ray radiography and computed tomography (CT) have been extensively used to detect the frequent presentations of pneumonia, pulmonary obstructions, and lung abnormalities [3]. At the same time, the role of nuclear imaging for detection and follow-up in patients with COVID-19 is gradually increasing.

In terms of treatment, antiviral therapy or antibody-based therapy is often used in the early stage of infection, while anti-inflammatory therapy is mainly used in the later stage to deal with the highly inflammatory state of patients [12].

Understanding the interaction between viruses and receptors is critical to understanding how to treat and prevent viruses. Due to the joint efforts of the whole world to fight against COVID-19, our understanding and management of COVID-19 have made significant progress, but further research is still needed. We are interested in the relationship between the regulation of ACE2 in vivo and the infection, progression, and transmission of COVID-19, since ACE2 is an important way for COVID-19 to enter the human body. At present, we mainly advance studies through three imaging modalities: fluorescence imaging, magnetic resonance imaging (MRI), and nuclear imaging.

## 3. Fluorescent Imaging

Fluorescent microscopy is a powerful and widely used optical technique for the visualization of cellular systems and processes in living cells or tissues. The basic principle involves the excitation of fluorescent molecules (be they proteins or dyes) by ultraviolet or visible light to enter a higher energy state. When the molecules release this absorbed energy, they emit longer wavelength (or lower energy) light that can be separated from the incident exciting light through the use of physical or digital filtration. This approach involves the balancing of spatial resolution, acquisition speed, and signal intensity, while avoiding photo-bleaching and photo-toxicity [13]. Fluorescent imaging has the advantages of very high spatial and temporal resolutions, sensitivity, and specificity. Together, a typical fluorescent microscopy system is capable of monitoring cellular dynamics in real time [14]. Although recent advances allow imaging on the nanometer scale, conventional techniques still offer high resolution at the sub-cellular scale [14]. Fluorescent imaging is well-established in the field of virology and was used to study coronavirus infections prior to SARS-CoV-2 (MERS-CoV and SARS-CoV) [15].

In SARS-CoV-2, ACE2 fluorescent imaging focuses on its interaction with the receptor-binding domain (RBD) on the viral spike protein, which is the initiating step of viral infection in the host cell and without which infection cannot occur [14,16]. The spike protein consists of two domains, the S1 and the S2, with the S1 responsible for ACE2 binding. The S1 domain is subdivided into the n-terminal domain, the RBD, and the c-terminal domain. In the “open” conformation, the receptor-binding site on the RBD is transiently exposed, allowing 17 amino acids on the binding site to interact with 20 amino acids on ACE2 [17,18,19]. RBD and ACE2 binding is similar between SARS-CoV and SARS-CoV-2. The RBD on SARS-CoV-2, however, has a higher affinity for ACE2 than RBD on SARS-CoV, although this affinity was found to be heterogeneous among viral strains [16]. Inhibition of this critical binding interaction may block SARS-CoV-2 infection and, therefore, it is an important target for antiviral therapeutics for COVID-19.

Thus far, the entire process of SARS-CoV-2 infection in living host cells has not been completely elucidated. To reveal key components in the infection cascade, fluorescent techniques enabling real-time imaging of living cells have been used by Miao et al. Here, the fate of the RBD–ACE2 complex in various ACE2-expressing cell lines has been determined using Halo and SNAP tags genetically encoded in the RBD and ACE2, respectively. Using structured illumination microscopy (SIM), they were able to monitor RBD binding to ACE2 and the subsequent endosomal internalization, leading finally to the degradation of the RBD and recycling of ACE2 to the cytoplasm [16]. SIM offers the advantage of being high resolution, while also being more compatible with living cell imaging than other super-resolution microscopy techniques [20]. Additionally, self-labeling proteins are highly specific, further enhancing the image quality [21]. Overall, their imaging results support their hypothesis for the fate of the RBD–ACE2 complex in living cells.

Fluorescent imaging of ACE2 has also been used for evaluating novel agents for inhibiting RBD binding to ACE2. Again, using SNAP- and Halo-tag-labeled ACE2 and RBD, respectively, inhibitors were screened on living cells, as the fluorescent signal from Halo–RBD will not stain cells docking when ACE2 is blocked [22]. A second method for screening ACE2 inhibitors involves the fluorescence signal generated by protein–protein interactions [23]. In this method, ACE2 and RBD are each fused to a different fragment of a fluorescent protein. When these two components are brought together in close proximity through protein–protein interactions, they reassemble and become fluorescent (Figure 1). Using this method, a large library of FDA-approved drugs and natural compounds were screened for their inhibition of ACE2 and RBD binding, quantified directly by the fluorescent signal.

It has been suggested that ACE2 dimerization is an important step in viral infection of a host cell; it has thus been presented as another target for antiviral therapeutics [24,25]. As previously mentioned, ACE2 may exist in either an open or closed state, where the open conformation is necessary for interaction with the spike protein. These states are also present in the proposed ACE2 homodimer, which is formed through ACE monomer interactions in the ferredoxin-like fold domain, which is referred to as the neck domain [26]. In the open homodimer state, ACE2 may bind two spike proteins simultaneously [25]. Zhu et. al. visualized this dimerization using a biomolecular fluorescence complementation assay (BiFC), where two fragments of YFP were fused with ACE2 monomers and co-expressed in HEK293T cells. In one study, ACE2 was expressed without the neck domain, while in the other, the neck domain was present. In a BiFC assay, YFP will only fluoresce when two fragments are brought together by protein–protein interaction. They observed that fluorescence was only observed in ACE2 containing the neck domain, confirming that dimerization is occurring and that the neck domain is important for this interaction. They also found that spike protein binding was not possible without the neck domain, suggesting that the neck domain, and potentially ACE2 dimerization, is essential for SARS-CoV-2 infection [26].

A major limitation of these fluorescent methods is that visible wavelength photons have limited penetration depth in tissues, and therefore, can be primarily utilized for in vitro cellular and biochemical studies. Additionally, fluorophores and the wavelengths used to excite them are damaging to cell viability and can affect cellular processes. This can be minimized through the choice of fluorophore and light exposure time [27]. Further, they rely on fragments of the spike protein instead of the SARS-CoV-2 virus itself, restricting broader generalizations and requiring evaluation using other methods. Despite this, fluorescent microscopy is a robust technology and allows visualization of cellular processes in real time, a limitation of many other techniques, making it a staple protein and ligand interaction imaging procedure, especially useful in the discovery phase.

## 4. MR Imaging

Magnetic resonance imaging is a non-invasive medical imaging technique that uses powerful magnets and radio waves to create detailed images of the internal structures of tissues and organisms. It relies on the principle of nuclear magnetic resonance, which involves the alignment of hydrogen protons within the body under a magnetic field and their subsequent response to radiofrequency pulses. MR has found extensive application in the medical field due to its ability to visualize various anatomical structures, such as the brain, spinal cord, muscles, and organs. It offers exceptional soft tissue contrast, enabling the identification of abnormalities, tumors, and injuries. Moreover, MR imaging is devoid of ionizing radiation, making it a safer alternative to other imaging techniques that involve X-rays that incur an absorbed dose that may be harmful in large quantities [28].

In the present context, MR imaging can be used to provide information about the consequences of ACE2 receptor binding, such as changes in tissue morphology, inflammation, or tissue damage associated with COVID-19 infection [29,30]. Since the ACE2 receptor is a protein, and the target of MRI detection is the proton signal in or around the tissue, MRI has not been used to directly image ACE2. However, Galisova et al. edited extracellular vesicles (EV) by genetic engineering to display RBD on its surface. As previously described, RBD can bind to ACE2, which is the initial step of COVID-19 infection. These EVs (RBD) were then labeled with superparamagnetic iron oxide nanoparticles to indirectly map ACE2 changes in vivo. The structure of EV(RBD) is very similar to coronavirus, but it is not contagious and has higher safety. Its ability to specifically target ACE2 was also demonstrated [31]. This strategy not only makes it possible to image ACE2 with MRI but also retains the possibility of using different nanocarriers including nano drug loading, providing the prospect of targeted drug delivery therapy (Figure 2).

## 5. Nuclear Medicine

### 5.1. Introduction to Nuclear Imaging

Nuclear imaging is an established medical imaging technique with many applications that include diagnostic cardiology and oncology. Conventional imaging methods including CT, MRI, and ultrasound provide detail from a structural or anatomical perspective. By contrast, nuclear imaging provides functional and metabolic information [32]. For example, the most widely used oncologic tracers are ^99m^Tc-labeled sulfur colloid to monitor lymphatic drainage around sites of breast or skin cancers, and ^18^F-FDG, a radioglucose analog that reports the in vivo consequences of the Warburg effect [33,34]. Here, high-energy gamma ray photons that are emitted by radioisotope-tagged molecules are sensitively detected. The reconstruction of tomographic images from these detected events provides a three-dimensional map of the activity concentration of these radiolabeled tracers. A distinguishing feature of nuclear imaging methods is a very high sensitivity for these measurements, into 63,108 the picomolar range within patients or preclinical models. Changes in physiological processes precede changes in anatomical perspectives, allowing nuclear imaging to assess organ and tissue function, blood flow, and cell activity. Such imaging information can be essential in diagnosis of cancer, heart disease, and neurological diseases. In addition, nuclear imaging is often combined with other techniques in order to provide anatomical information, resulting in fusion imaging that includes PET/CT, SPECT/CT, and PET/MRI.

Nuclear imaging provides information about the patient using the distribution of radioactive tracers injected into the patient. The most common nuclear imaging modalities are positron emission tomography (PET) and single-photon emission computed tomography (SPECT). PET imaging involves the use of positron-emitting radiotracers; in which coincident gamma rays are emitted when these positrons annihilate with surrounding electrons [35]. This allows the measurement of metabolic and biochemical processes. SPECT imaging, on the other hand, utilizes radiotracers that emit gamma ray photons directly. These photons are detected using a gamma camera that orbits the patient to capture multiple views [36].

### 5.2. ACE2 Imaging with DX600

In 2003, Huang et al. evaluated an M13 filamentous phage screen against the ACE2 receptor and, using this, discovered a novel inhibitor, DX600 (Ki = 2.8 nM). The key domain for the combination of DX600 and ACE2 is CSPLRYYPWWKC [37]. DX600 also contains an intramolecular disulfide bond, which can form a strong and stable complex with the amino acid residues on the ACE2 protein and can also better fit the active site or binding pocket of ACE2. These features ensure that DX600 can bind ACE2 with high affinity, high stability, and specificity. This gives DX600 excellent potential as a molecular probe to localize ACE2 expression in vivo in a non-invasive manner.

At this time, several DX600 structure-based derivatized radiotracers have been evaluated for nuclear imaging in preclinical and clinical settings. These have been evaluated using a variety of chelates (to stably complex with radiometals) and using a range of positron decay nuclides (^18^F, ^68^Ga, ^64^Cu, ^89^Zr). A variety of ACE2 tracers have been used to evaluate the expression of ACE2 in different organs throughout the body during COVID-19 from different perspectives, and several of them have been evaluated in clinical trials see Table 1, [38,39].

The first attempts at DX600-based radiotracers involved modifying the N-terminus of DX600 by conjugating different radiometal chelators. Two chelators and isotopes were evaluated in one of the initial studies, NODAGA (^64^Cu-HZ20) for ^64^Cu labeling and DOTA (^68^Ga-HZ20) for ^68^Ga labeling. Both ^64^Cu-HZ20 and ^68^Ga-HZ20 showed a high accumulation in a xenograft model with high ACE2 expression in mice and exhibited ideal pharmacokinetic properties in a human ACE2-expressing genetically engineered mouse model.

Further translation has focused on ^68^Ga-HZ20, which has been used in a clinical study (NCT04422457). Here, volunteers and a patient who had recovered from COVID-19 were evaluated for organ uptake and dosimetry. The conclusions drawn from the regions of uptake were largely the same as those in the Human Protein Atlas database [40]. One notable exception was that ^68^Ga-HZ20 tracer accumulation was observed in the female mammary gland, whereas the database reported no ACE2 expression in this organ. Compared with the control group of healthy volunteers, tracer accumulation in the nasal mucosa (patients with partial or complete anosmia), digestive tract (patients with diarrhea or vomiting), testis, corpus luteum (affecting the patient’s reproductive function), and kidney (patients may experience acute kidney injury) was high (Figure 3). However, there was lower accumulation in the heart and lungs, which may have been due to differences in the expression of ACE2 in these organs, the reasons for which have not been elucidated in detail [38]. Tracer accumulation in the recovered patient suggests that ACE2 upregulation is associated with post-COVID-19 physiological changes. Although the first study had some limitations, such as a small sample size of patients, it succeeded in non-invasively evaluating the relationship between COVID-19 symptoms and ACE2 expression in patients from a new perspective.

Subsequently, further research on these DX600-based tracers has been carried out from two aspects. On the one hand, the evaluation of different variants of the structure has been performed. Another aspect is to further conduct more in-depth research on COVID-19 from different angles through DX600-based tracers.

First, starting from the structure of the tracer, Parker et al. conducted a study based on the structure of radiolabeled DX600 to prove the importance of the disulfide bond in DX600. Tracers that lose their disulfide bonds become linear structures and have greatly reduced activity toward ACE2. Furthermore, they confirmed that the addition of different linkers between the DX600 end and the metal chelator had no effect on the ability of the DX600 tracer to localize ACE2. The binding affinities of a series of tracers developed by them are similar to or even lower than that of DX600 itself, suggesting great potential for follow-up research [41]. These also further confirmed the results from the previously described study [38]. Use of an Al^18^F-labeled DX600 BCH for clinical utilization has also been started (NCT04542863) [39]. The motivation for this approach is the higher resolution afforded by the emission energy profile of the positron from Fluorine-18, relative to Gallium-68. In additional work to study alternative radionuclides and imaging technologies, Bayer et al. visualized the distribution of ACE2 in mice using SPECT/CT imaging using ^67^Ga-labeled HZ20 and its analogs [42]. The conclusions from these studies were confirmed with PET, revealing the potential of SPECT in ACE2 imaging.

A variety of radiolabeled DX600 and its analogs (such as ^68^Ga-cyc-DX600) are currently being used in the research of COVID-19 symptoms, prevention, transmission, and recovery in both humans and in model organisms [43]. Among them, ^68^Ga-cyc-DX600 was used to observe the effectiveness of the COVID-19 vaccine in a rabbit model. The long observation window provided by ^68^Ga enables one to monitor the fluctuation of ACE2 in specific organs after vaccination non-invasively [44]. This may guide the development of vaccines, helping to evaluate their effectiveness and potentially shortening the time for their approval. Similar tracers have also been used to study the distribution of ACE2 in pigeons [45]. As 75% of the key sites of ACE2 in pigeons overlap with humans, they may serve as a potential intermediate host to complete the spread in cities. Given the concerns related to mammalian viral transmission in avian species, this study also helps to understand COVID-19 from the perspectives of communication and viral evolution [46,47].

### 5.3. ACE2 Imaging with RBD

This method for localizing ACE2 is similar to the strategy used for fluorescence imaging, utilizing the receptor-binding domain of the SARS-CoV-2 virus itself to localize to sites of expression. These imaging modalities all rely on the binding of RBD to ACE2 since the combination of RBD and ACE2 is the only way for the virus to infect the body. In the direction of nuclear imaging, the first attempt to use this imaging strategy was by Li et al. Non-invasive imaging using PET with an ^124^I-RBD tracer revealed a significant accumulation in high ACE2-expressing organs in HCC-HepG2 mice [48]. In addition, Li et al. used virological methods to study the fluctuation of ACE2 expression in vivo after infection with COVID-19 by using enhanced fluorescent protein and radioactively labeled pseudoviruses [49]. Radiolabeled nanoparticles have also been used to reveal the distribution of RBD in vivo [50]. Thus, targeting RBD is also a promising method to indirectly image the distribution of ACE2, opening new ideas for the prevention and treatment of COVID-19.

### 5.4. Nuclear Imaging Shortcomings

The current use of nuclear imaging to study the relationship between COVID-19 and ACE2 is an active and promising field. However, there are inevitably still some limitations. The first is the small number of samples involved in these studies to date. This may be a limitation imposed by the PET/SPECT imaging technique itself, as academic medical centers are typically the locations for development of novel radiotracers. At this time, dissemination of these probes to primary and tertiary medical sites has been limited. Nuclear imaging also relies on ionizing radiation as a means of generating a signal, which must be carefully monitored for patient and health-care worker safety. In addition, the use of nuclear imaging to study the distribution of ACE2 is dependent on the expression of ACE2. However, in the course of the disease, ACE2 may be dysregulated, and the expression level of ACE2 in different patients may be different. The expression of ACE2 in the body is also dynamic. Although different nuclides can provide different time windows to monitor the fluctuation of ACE2 in the body, the radionuclides and tracers used to date provide a snapshot into the specific time segments of this biology. Further work will need to be performed to further develop and evaluate DX600- or RBD-based tracers to study the dynamic distribution of ACE2 in vivo in response to infection and vaccination.

## 6. ACE2 Imaging in Cancerous Tissues

Beyond its physiological and infectious disease potential, upregulation of ACE2 has been demonstrated in many tumors [6]. Overall, ACE2 is a double-edged sword for tumor growth and formation. On the one hand, ACE2 has been described as inhibitory, reducing angiogenesis during tumor growth by reducing cell proliferation and promoting cell death [51,52]. Conversely, data also shows that dysregulation of ACE2 may promote tumor growth. The expression or mutation of ACE2 can be detected on the surface of most tumors, and the expression of ACE2 itself is also affected by the tumor microenvironment, which in turn affects the development of cancer [6,53,54]. This opens up the possibility of imaging ACE2-expressing tumors for the targeting of ACE2 to monitor tumor formation and development.

Above, we detailed the use of nuclear imaging to detect ACE2-expressing tumors in pilot and confirmatory studies in the translation of DX600-based tracers. Specifically, these were conducted to confirm the specificity of radiotracers such as ^68^Ga-cyc-DX600 to ACE2 [38]. In further clinical work, Ren et al. compared the uptake of ^68^Ga-cyc-DX600 with ^18^F-FDG in patients with a variety of tumor types and diseases. They showed that ^68^Ga-cyc-DX600 as an ACE2-specific imaging tracer could complement ^18^F-FDG to provide a more comprehensive image. For example, in a clinical examination of a patient with cirrhosis, ^18^F-FDG PET/MR showed increased uptake in distal esophageal lymph nodes, which may suggest metastasis, but no corresponding increase in uptake by ^68^Ga-cyc-DX600; however, ^68^Ga-cyc-DX600 showed hyperdense nodules in the lungs, which were not ^18^F-FDG avid (Figure 4). As ^68^Ga has a relatively short half-life (68 min), it has the potential to be used in the clinical setting for same-day patient imaging without onerous changes to present workflows [55]. The targeting of tumors for therapy with beta-particle-emitting radionuclides for theranostics has also been proposed. Here, labeling of HZ20 with ^177^Lu via the DOTA chelator yielded a probe that can provide information on distribution and uptake using SPECT, as well as deliver significant doses of targeted radiotherapy for an anticancer effect. The initial study indicates that ^177^Lu-HZ20 has the potential to be a comprehensive probe for diagnosis and therapy [56].

At present, with the deepening of COVID-19 research, research in this field has become a hot spot, but it is still in its infancy. Therefore, locating ACE2 in the tumor environment is of great significance for understanding the relationship between them, which is helpful for the prevention and prognosis of tumors, and can also evaluate the efficacy of anticancer therapeutics, laying the foundation for new imaging methods and diagnosis.

## 7. ACE2 Imaging in Liver Disease

ACE2 is expressed to varying degrees in almost every organ system [57]. Abnormally elevated enzyme levels in the liver in patients with no prior liver disease is a common manifestation of SARS-CoV-2 infection, where ACE2 is moderately expressed [7,58]. ACE2 expression in the liver has, therefore, become a focus area of interest in understanding chronic liver disease.

Chronic liver disease is a significant cause of childhood morbidity and mortality [59,60]. It involves the gradual destruction of the liver, eventually leading to cirrhosis, fibrosis, and other complications [60]. Previous data have reported that liver tissue from adult patients with liver disease expressed higher levels of ACE2 than normal liver tissues, but these results are not well validated in pediatric patients [58]. Recently, Stevens et al. examined ACE2 expression in pediatric patients with chronic, immune-mediated liver disease using immunofluorescence. In immunofluorescence, an antibody targeting a specific protein, in this case, ACE2, is conjugated to a fluorophore [61]. A fixed tissue sample can then be evaluated for target protein expression using the fluorescent signal. Stevens et al. developed a method to identify different cell types, as well as to segment the cytosol and the nucleus. In this way, they were able to quantify the fluorescent activity in different cells and cell regions. The study found that ACE2 may be overexpressed in pediatric liver disease, as well as distributed differently among cells in the hepatic tissue. This initial study requires further expansion but provides indicators that ACE2 expression in liver disease may be a valid target for imaging in order to detect and monitor the progression of this disease (Figure 5).

## 8. Conclusions

Various imaging modalities for detecting ACE2 are playing an expanding role in guiding the detection and treatment of various ACE2-related diseases. Nuclear imaging studies in particular provide a translational pathway to study the distribution of this receptor in the contexts of infectious disease, cancer, and potentially in liver disease. The limitations of nuclear imaging that must be considered include the use of ionizing radiation, and the limited scope of studies to date. There is also significant potential in the application of both imaging and radiotherapy in the form of theranostics to target sites of disease overexpressing ACE2. Complementing these approaches, fluorescent imaging provides the advantages of imaging living cells at a high resolution, permitting real-time monitoring of cellular dynamics and processes; however, depth imaging will be challenging. MRI can utilize the combination of coronavirus mimetics and nanoparticles to target ACE2 expression in vivo.

These imaging tools can assess ACE2 fluctuations in the body in normal physiological states and in response to the presence of disease. Imaging tools for the sensitive and quantitative definition of ACE2 can assist in the development of therapeutic methods. This has the power to inform a better understanding of various diseases, especially during the COVID-19 pandemic, and can inform new ways of prevention, detection, and treatment. Similarly, for cancer and liver diseases, the information provided by these imaging modalities can also provide new therapeutic and diagnostic methods. The imaging characteristics of different imaging methods and the development of subsequent treatment methods are of continuous research interest; they can provide physicians with more patient information and provide patients with more treatment options, which will lead to better treatment results.

The ongoing research may also have some limitations, such as how to translate the results from the bench to the bedside and how to expand the sample size in clinical trials to make the conclusions more solid. Regardless, we can see that targeting ACE2 imaging opens new windows for understanding the prevention, treatment, and transmission of multiple diseases.

## Figures and Tables

**Figure 1 viruses-15-01982-f001:**
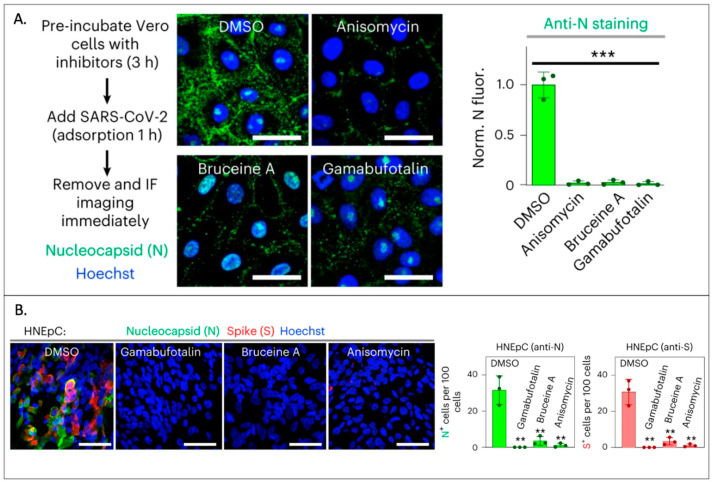
Fluorescent imaging of ACE2 to evaluate ACE2 inhibitors. (**A**) In the presence of inhibitors, VERO-E6 cells demonstrated low fluorescence when stained with anti-nucleocapsid antibodies (green). (**B**) Primary human nasal epithelial cells (HNEpC) similarly displayed low fluorescence in the presence of the same inhibitors when stained with anti-nucleocapsid (green) or anti-spike (red) antibodies [23]. ** *p*  <  0.01, *** *p*  <  0.001.

**Figure 2 viruses-15-01982-f002:**
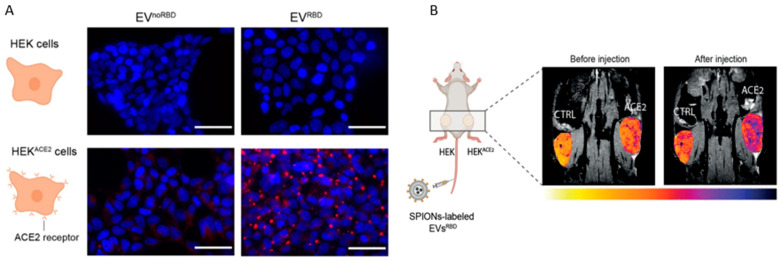
MR imaging ACE2 with EV. (**A**) Fluorescent images of HEK293 cells and HEK293 cells expressing ACE2 receptors, incubated with red fluorescently labeled EV^noRBD^ and EV^RBD^. Cell nuclei staining was carried out by DAPI, which shows blue in the image. (**B**) MR images obtained before magnetically labeled EVs^RBD^ injection and 4 h after injection. Tumors: control (CTRL, left), HEK^ACE2^(ACE2, right) [31].

**Figure 3 viruses-15-01982-f003:**
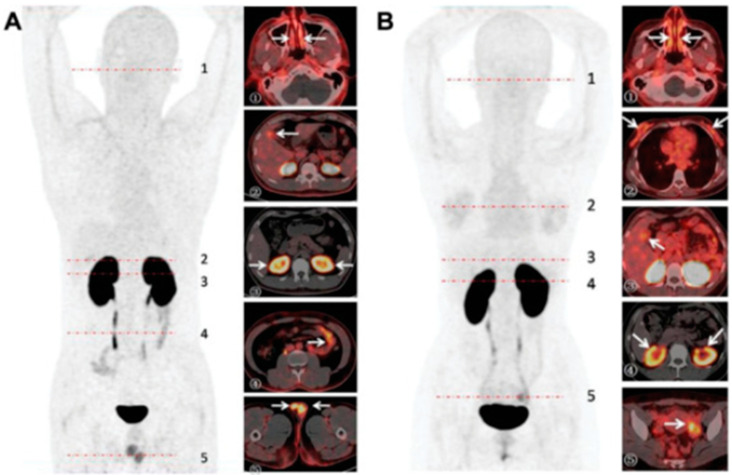
PET imaging of 68Ga-HZ20 in male and female volunteers. Radioactivity uptake in two COVID-19 patient volunteers at 90 min post-injection, shown as MIP and transverse images. (**A**) Male (38 years old), the nasal mucosa (1), gallbladder (2), and small intestine (4) showed moderate radioactivity accumulation, and the renal cortex (3) and testis (5) showed high accumulation. (**B**) Female (34 years old), the nasal mucosa (1), breast (2), and gallbladder (3) showed moderate accumulation, and the renal cortex (4) and the corpus luteum of the left ovary (5) showed high accumulation [38].

**Figure 4 viruses-15-01982-f004:**
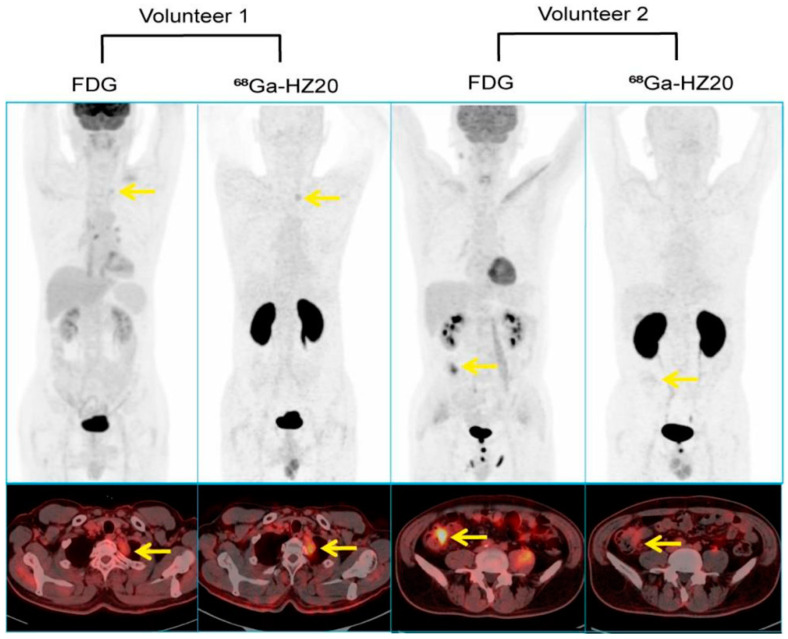
Comparison between ^18^F-FDG and ^68^Ga-HZ20 PET/CT scans. In volunteer 1, ^68^Ga-HZ20 showed a higher SUVmean value than ^18^F-FDG; in volunteer 2, corresponding SUVmean values are opposite, yellow arrows indicate lesions. ^18^F-FDG shows thickening of the wall of the ascending colon, while ^68^Ga-HZ20 only shows low uptake. The patient underwent laparoscopic right hemicolectomy and pathological examination, the pathological assessment was poorly differentiated colon adenocarcinoma; IHC analysis of the primary lesion showed low ACE2 expression (<5%) [56].

**Figure 5 viruses-15-01982-f005:**
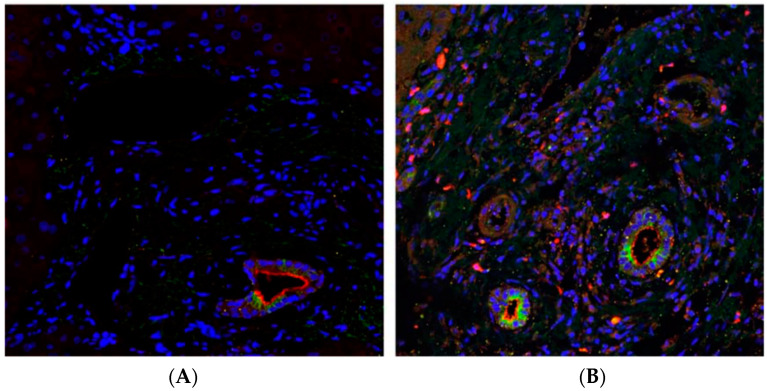
ACE2 staining in the liver. ACE2 immunofluorescence is shown in red. CK-19, a cholangiocyte marker, is shown by green immunofluorescence. Blue staining shows cell nuclei. (**A**) ACE2 immunofluorescence seen in adult hepatic tissue. The authors found ACE2 localization primarily on the apical membrane of cholangiocytes. (**B**) ACE2 immunofluorescence in hepatic tissue from a pediatric patient with primary sclerosing cholangitis. ACE2 expression is found throughout many different cell types, in contrast to the adult hepatic tissue [7].

**Table 1 viruses-15-01982-t001:** Clinical trial of selected tracers for molecular nuclear imaging of ACE2.

CTID	Abbreviated Titles	Imaging Modality	Purpose	Compound	Patients	Status
NCT04422457	Specific Molecular Imaging of DX600 Labeled by PET Radionuclide Targeting ACE2 in Patients	PET/CT	Diagnostic	^68^Ga-DX600	30	Completed
NCT04542863	The Molecular Imaging Research of F-18-Labeled DX600 PET Probe	PET/CT	Diagnostic	^18^F-DX600	100	Recruiting

## Data Availability

This review article references all of the information present and may be available through the original authors.

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
