# Peer review of "Molecular Imaging of ACE2 Expression in Infectious Disease and Cancer"

_viruses, 2023, doi:10.3390/v15101982_

Round 1

Reviewer 1 Report

The paper "Molecular Imaging of ACE2 Expression in Infectious Disease and Cancer" by Li and colleagues is an interesting, albeit a bit short, review on the existing tools and techniques for ACE2 investigation. It is an interesting read, and I thank the authors for having put the effort in it. Below, my minor points and observations.

Line 71: the notion that SARS-CoV-2 was first reported in China in late February 2019 is a bit strange, given the fact that the first molecular identification of the virus dates in December 2019. Can the authors correct this date, or, if they have evidence of its correctness, provide a reference for it?

In general, since the authors provide many references to animal ACE2, it would be beneficial to the manuscript if they could highlight (also quantitatevely) the high structural similarity between human and rodent ACE2 (for example).

What are the chances of antibody-related imaging of ACE2 be contaminated by similar epitopes from other proteins (for example, TMPRSS2 or ACE1), and how should the readers be aware of potential false positive immunochemical identification of ACE2?

Author Response

We would like to express our sincere gratitude to you and the esteemed reviewers for dedicating your time and expertise to evaluate our manuscript. We respond to the comments by the author through corrections and changes throughout the manuscript. 

For reviewer #1, we have numbered the comments and compiled our point-by-point response here:

R1.Q1)  Can the authors correct this date, or, if they have evidence of its correctness, provide a reference for it?

Response: We thank the reviewer for catching this error. We have corrected this typo.

R1.Q2) In general, since the authors provide many references to animal ACE2, it would be beneficial to the manuscript if they could highlight (also quantitatevely) the high structural similarity between human and rodent ACE2 (for example).

Response: To the introduction we have added text and references regarding the inability of ACE2 to recognize the Spike protein of the virus. We write: "The majority of animal models used to study infectious disease, and COVID-19, are mouse in origin. Despite a high degree of conservation, murine ACE2 differs from the human protein and does not bind the SARS-CoV-2 spike (S) protein. Therefore, transgenic human-ACE2 mice have been used in preclinical imaging agent development, and in tumor xenograft models (60,61)."

R1.Q3) What are the chances of antibody-related imaging of ACE2 be contaminated by similar epitopes from other proteins (for example, TMPRSS2 or ACE1), and how should the readers be aware of potential false positive immunochemical identification of ACE2?

Response: There has been substantial effort in the biomedical community to validate the targeting tools with a special emphasis on specificity. Many of these targeting tools are the basis for the imaging agents reviewed in this manuscript. We are not aware of reports of specific off target binding of antibodies, peptides or genetic fluorochrome constructs that bind to other enzymes involved in SARS-CoV-2 pathology. This includes a lack of reactivity of DX600 peptide to ACE1. 

Reviewer 2 Report

The author provides a compressive overview of ACE2 visualization tech, with a clear logical structure and crystal expression. This work could benefit experts in pharmacology and pathology. I strongly recommend its publication. There is a quick question: ACE2 dimerization is crucial in the disease process. Are there any imaging methods available to visually analyze whether it is dimerized?

Author Response

We thank the reviewer for their kind reception of our review article. The reviewer has asked a question regarding the visualizing the dimerization of the ACE2 molecule. In our original submission, work using  Bimolecular fluorescence complementation (BiFC) was overlooked. We have now included a paragraph describing an important model system using a fluorescent reporter to study dimerization of ACE2.

Reviewer 3 Report

In this review, ‘Molecular Imaging of ACE2 Expression in Infectious Disease 2 and Cancer’ the authors recognize the state of the art of ACE2 in health and diseases by focusing on imaging tools such as fluorescence, Magnetic Resonance Imaging (MRI), and Nuclear Imaging.

Data interpretation is well discussed, and the message is clearly stated.

The literature cited is relevant, and the author aims to introduce their own style.

The figures are clear and informative and graphically depict the factors that influence their application.

The review may be accepted for publication in this journal without revision.

Author Response

We thank the reviewer for their kind comments on the review.